# Effects of Hemorrhage on Hematopoietic Cell Depletion after a Combined Injury with Radiation: Role of White Blood Cells and Red Blood Cells as Biomarkers

**DOI:** 10.3390/ijms25052988

**Published:** 2024-03-04

**Authors:** Juliann G. Kiang, Akeylah K. Woods, Georgetta Cannon

**Affiliations:** 1Armed Forces Radiobiology Research Institute, Uniformed Services University of the Health Sciences, 4555 South Palmer Road, Building 42, Bethesda, MD 20889-5648, USA; akeylah.woods@usuhs.edu (A.K.W.); georgetta.cannon@usuhs.edu (G.C.); 2Department of Medicine, Uniformed Services University of the Health Sciences, Bethesda, MD 20814, USA; 3Department of Pharmacology and Molecular Therapeutics, Uniformed Services University of the Health Sciences, Bethesda, MD 20814, USA

**Keywords:** mouse, radiation, hemorrhage, white blood cell, red blood cell, platelet

## Abstract

Combined radiation with hemorrhage (combined injury, CI) exacerbates hematopoietic acute radiation syndrome and mortality compared to radiation alone (RI). We evaluated the effects of RI or CI on blood cell depletion as a biomarker to differentiate the two. Male CD2F1 mice were exposed to 8.75 Gy γ-radiation (^60^Co). Within 2 h of RI, animals were bled under anesthesia 0% (RI) or 20% (CI) of total blood volume. Blood samples were collected at 4–5 h and days 1, 2, 3, 7, and 15 after RI. CI decreased WBC at 4–5 h and continued to decrease it until day 3; counts then stayed at the nadir up to day 15. CI decreased neutrophils, lymphocytes, monocytes, eosinophils, and basophils more than RI on day 1 or day 2. CI decreased RBCs, hemoglobin, and hematocrit on days 7 and 15 more than RI, whereas hemorrhage alone returned to the baseline on days 7 and 15. RBCs depleted after CI faster than post-RI. Hemorrhage alone increased platelet counts on days 2, 3, and 7, which returned to the baseline on day 15. Our data suggest that WBC depletion may be a potential biomarker within 2 days post-RI and post-CI and RBC depletion after 3 days post-RI and post-CI. For hemorrhage alone, neutrophil counts at 4–5 h and platelets for day 2 through day 7 can be used as a tool for confirmation.

## 1. Introduction

Victims that are exposed to radiation are frequently exposed to other trauma, including, for instance, wounds, brain injuries, bone fractures, burns, or hemorrhage (Hemo). After the bombings at Hiroshima and Nagasaki, appreciable cases of these combined injuries were detected [1,2]. Furthermore, many victims at the Chernobyl reactor meltdown were exposed to radiation combined with another trauma [3]. In rodent models, combined radiation and trauma injury (CI) increased organ damage and fatality after otherwise nonlethal irradiation [4,5,6,7]. Secondary consequences to survivors of these combined injuries are known to show exacerbated acute radiation syndrome (ARS), including enteropathy (GI-ARS) associated with hematopoietic syndrome (H-ARS) [8,9]. Most importantly, the bone and bone marrow damage caused by CI is more than that caused by either single trauma alone [10,11]. The acute loss of bone mass after CI occurs because of a rapid rise in activation and activity of osteoclasts [10] in conjunction with attenuated activity of osteoblasts [12], resulting in decreases in the formation and volume of bone [13]. Therefore, under a nuclear attack, nuclear accident, or exposure to a radioactive dispersal device (RDD), the unfavorable decline in bone formation and volume can cause a long-term health impact in first responders and survivors [14,15].

Either Hemo or radiation injury (RI) can result in similar outcomes depending on the % of blood loss or the radiation dose. Hemo at 40% of total blood volume, which was classified as level 4 damage [16], increases the concentrations of interleukin-10 (IL-10) and tumor necrotic factor-α (TNF-α) in blood, activates nuclear factor-keppaB (NF-κB) and inducible nitric oxide synthase (iNOS) expression in the murine small intestine, and increases cell apoptosis in various murine tissues [17,18,19,20]. Likewise, sub-lethal ionizing radiation also increases these parameters [4] in addition to initiating deleterious hematopoietic changes [8]. Even though Hemo at 20% of total blood volume was classified as damage level 2 and causing no harm, [16], when it occurred following radiation exposure, it resulted in a 25% increase in mortality above that of radiation alone (50% mortality) or Hemo alone (0% mortality) [6]. When the Hemo is internal, a combined radiation + Hemo condition may be difficult to diagnose. If blood cell depletion could be used as an indicator of the synergistic effects of these two traumas, it would allow for the diagnosis of combined radiation–Hemo injury, leading to more aggressive treatment strategies than a single injury would require and thus better survival under scenarios of nuclear accidents possibly involving Hemo. 

Thus, the aim of this study was to evaluate the effects of radiation injury (RI) followed by non-lethal Hemo (i.e., CI) on blood cell depletion to develop indicators to identify Hemo occurrence concurrently with irradiation. We hypothesize that the combined trauma of Hemo following radiation would be more detrimental to complete blood count (CBC) depletion than either assault alone. Mice were used to test the hypothesis, because studies with a whole animal include interaction among organs and molecular components of transcription factors, cytokines, and microRNA (miR). Our data support the contention that changes in CBC depletion derived from CI mice compared to single-injury mice can act as the needed indicators.

## 2. Results

### 2.1. CI Reduces White Blood Cells (WBCs) More Than RI Alone on Day 1 and Day 2

Whole-body irradiation at a high radiation dose reduced white blood cell counts (WBCs) immediately, a hallmark of RI [8]. Figure 1a depicts that 20% hemorrhage alone did not alter the total WBCs at 4–5 h post the initial bleed, but WBCs declined on day 1 in comparison to the sham groups. However, WBCs returned to the baseline quickly on days 2, 3, 7, and 15. RI and CI decreased WBCs at 4–5 h. CI appeared to further decrease WBCs on day 1 and day 2 (Figure 1a,b), but both CI and RI groups stayed at nadir on days 3, 7, and 15, with similar values (Figure 1a). To understand which subgroups of blood cells lead to these further decreases, WBC differentiation was conducted.

### 2.2. CI-Induced Neutrophil Depletion Is Greater Than That of RI Alone on Day 2

The whole-body irradiation-induced WBC reduction is evidently due to neutropenia and lymphocytopenia [8]. Figure 1c shows that neutrophils were increased at 4–5 h post-Hemo alone and then returned to the basal levels on days 1, 2, 3, 7, and 15. On day 1, neutrophil counts in RI and CI groups remained like the sham and the Hemo groups. However, on day 2, RI and CI reduced neutrophil counts, and the CI-induced reduction was more than the RI-induced one (Figure 1c,d). Afterwards, RI and CI neutrophil counts remain at their nadir on days 3, 7, and 15, with similar values (Figure 1c). 

### 2.3. CI-Induced Lymphocyte Depletion Is More Than RI Alone on Day 2

Unlike neutrophils, Figure 2a and b show that Hemo significantly reduced lymphocyte counts at 4–5 h post the initial bleed and at day 1 after Hemo, but lymphocyte counts returned to the basal level starting day 2 through day 15. RI and CI vastly decreased lymphocyte counts. At day 2 post-RI and post-CI, CI briefly decreased lymphocyte counts further. Then, RI and CI lymphocyte counts remained at their nadir on days 3, 7, and 15, with similar counts (Figure 2a). 

### 2.4. CI-Induced Monocyte Depletion Is More Than RI Alone on Day 1

Figure 2c,d show that Hemo alone monocyte counts were slightly higher than those of the sham group at 4–5 h post the initial bleed, followed by a slower increase on day 1 and then a full recovery. RI and CI monocyte counts were significantly reduced beginning at 4–5 h post-RI and CI, continued to drop on day 1 and day 2, and then stayed at nadir from day 3 to day 15. CI reduced monocyte counts more than RI only on day 1.

### 2.5. CI-Induced Eosinophil Depletion Is Greater Than That of RI Alone on Day 2

Eosinophils are known to be related to allergy and the suppression of RI-induced small intestinal fibrosis [21]. The counts for the sham and Hemo groups were indistinguishable, except on day 7, when the eosinophil counts were lower in the Hemo group than in the sham group (Figure 3a,b). The counts recovered on day 15 (Figure 3a). Eosinophil counts were decreased by RI and CI beginning at 4–5 h and through days 1–15 (Figure 3a). On day 2, CI reduced eosinophil counts more than RI (Figure 3b). 

### 2.6. CI-Induced Basophil Depletion Is Greater Than That of RI Alone on Day 2

Basophils have emerged as a small but potent subpopulation of leukocytes capable of bridging innate and adaptive immunity [22]. Basophil counts were decreased at 4–5 h after Hemo alone but recovered back to baseline on days 1, 2, 3, 7, and 15 (Figure 3c). RI and CI reduced basophil counts at 4–5 h, which was even lower than the Hemo group, and thereafter to day 15 (Figure 3c). However, on day 2, CI depleted basophil counts more than RI (Figure 3d). 

### 2.7. CI Reduces Red Blood Cell Counts (RBCs) More Than RI Alone on Days 3, 7, and 15

Unlike WBCs, RBCs were reduced at 4–5 h and thereafter on days 1 and 2 post-Hemo. On day 3, RBCs were recovering and back to normal on day 15. The RI-induced RBC depletion began to show on day 3 and continued decreasing on day 15. In contrast to RI, the CI-induced decrease in RBCs appeared at 4–5 h, it was sustained at this low level to day 3, and then it decreased further on days 7 and 15 (Figure 4a,b). Figure 4c shows that CI induced a greater RBC depletion than RI through every time point, as evidenced by the slope of CI (4.7266) versus the slope of RI (3.4373, *p* < 0.05). 

### 2.8. CI Reduces Hemoglobin More Than RI Alone on Days 3, 7, and 15

Similar to RBCs, hemoglobin levels were reduced at 4–5 h and thereafter on days 1 to 3 post-Hemo. On day 7, hemoglobin levels were fully recovered and remained normal on day 15. The RI-induced hemoglobin began to show on day 2 and continued decreasing to day 15. In contrast to RI, the CI-induced decrease in hemoglobin appeared at 4–5 h, was sustained at this low level to day 3, and then further decreased on days 7 and 15 (Figure 5a,b). Figure 5c shows that CI induced greater reductions in hemoglobin than RI through every time point, as evidenced by the slope of CI (4.6786) versus the slope of RI (3.7274, *p* < 0.05). 

### 2.9. CI Reduces Hematocrit More Than RI Alone on Days 3, 7, and 15

Similar to RBCs and hemoglobin, hematocrit readings were reduced at 4–5 h and thereafter on days 1 to 2 post-Hemo alone. On day 3, hematocrit readings were partially recovered and fully normal on days 7 and 15. The RI-induced hemoglobin drop began to show on day 3 and continued decreasing to day 15. In contrast to RI, the CI-induced decrease in hematocrit readings appeared at 4–5 h, it was sustained at this low reading to day 3, and then it decreased on days 7 and 15 (Figure 6a,b). Figure 6c shows that CI induced a sharper decline in hematocrit readings than RI through every time point, as evidenced by the slope of CI (4.6737) versus the slope of RI (3.6985, *p* < 0.05). 

### 2.10. Hemo Drives Increases in Platelet Counts on Days 2, 3, and 7, but CI Does Not Reduce Platelet Counts More Than RI Alone on Days 3, 7, and 15

Unlike WBCs and RBCs, platelets were not reduced at 4–5 h or thereafter on day 1 post-Hemo. Hemo increased platelets on days 2, 3, and 7 and returned platelets to the sham level on day 15. RI and CI did not decrease platelet counts until day 7, and these continued to decline on day 15. The decreases driven by either RI or CI were similar (Figure 7a,b) and confirmed by Figure 7c. 

## 3. Discussion

High-dose radiation exposure accompanied with Hemo is one of the common CIs expected after radiological exposure [23]. Herein, we report that WBCs are early indicators (within 3 days) and that RBCs are later indicators (later than 3 days and up to 15 days) for radiation victims with Hemo. In contrast to WBCs and RBCs, no differential platelet depletion was found between RI and CI. This platelet result is inconsistent with what was found in combined radiation injury with wound trauma [8], suggesting that hemorrhage trauma is less impactful to platelets compared with wounding.

Hemo at 20% of total blood volume was classified as not requiring fluid resuscitation [16]. Hemo significantly increased neutrophils (Figure 1c) and platelets (Figure 7a,b); the phenomenon was like that after wounding because of the need of wound healing and protection against bacterial infection [4].

RI is known to result in endothelium injury [24,25] that is exacerbated by CI due to the upregulation of both EPO and HIF-1α [6]. The important issue is to differentiate between the RI victims and CI victims, particularly those with internal bleedings. Our data suggest differential blood cell counts can be used to determine this depending on the time increments after the incident. As shown in Table 1, if the blood samples were collected 4–5 h after the incident, Hemo victims showed increases in neutrophils and monocytes and decreases in other WBCs, RBCs, and platelets; RI victims showed decreases in WBCs but not in RBCs and platelets; and CI victims showed increases in neutrophils and decreases in RBCs and platelets. On day 1, Hemo victims showed normal neutrophils, decreased RBCs, and normal platelets; RI victims showed decreased WBCs and normal RBCs and platelets; and CI victims showed further decreased monocytes and normal RBCs and platelets. On day 2, Hemo victims fully recovered WBCs, but RBCs remained low and platelets were increased above the baseline; RI victims showed decreased WBCs, normal RBCs, and decreased platelets. CI victims showed further decreases in WBCs and RBCs but normal platelets. On day 3, Hemo victims showed normal WBCs and RBCs but increased platelets above the baseline; RI victims showed decreased WBCs and RBCs and normal platelets; and CI showed the same decreases in WBCs and further decreases in RBCs but normal platelets. Like on day 3, on day 7, Hemo victims showed normal WBCs and RBCs but increased platelets; RI showed decreased WBCs and RBCs but increases in platelets above the basal level; and CI victims appeared to show decreased WBCs and further declined RBCs and platelets. On day 15, Hemo victims displayed normal CBCs; and RI victims appeared to show low WBCs and platelets and further lower RBCs. Collectively, the data suggest that on day 1, monocyte counts can distinguish RI from CI, whereas on day 2, all WBC subgroups can separate RI from CI. From day 3 and on, RI and CI victims could be separated by the severity of RBC depletion. 

RI and CI induced CBC depletion in different cell types at different time points. This laboratory previously reported that RI activated NF-κB, leading to increased iNOS and cytokine expression and thus to caspase-3-dependent apoptosis [6]. Subsequently, bone marrow (BM) cellularity was decreased. CI further enhanced the NF-κB activation, cytokine production, iNOS upregulation, and caspase-3 activation [4] and resulted in an additional decrease in BM cellularity [6]. There are also differences found in gastrointestinal ARS between RI and CI [26]. The opportunity for finding useful endpoints from gastrointestinal ARS cannot be excluded and should be explored.

Hemorrhage alone increased neutrophil counts at 4–5 h and platelet counts at day 2 through day 7. This may be due to platelet–neutrophil interactions. Platelets express P-selectin on their surface, which interacts with P-selectin granulocyte ligand 1 (PSGL1) of neutrophils [27,28] to form a complex after RI and CI. This leads to vessel injury and/or endothelial activation [29]. Moreover, it is known that megakaryocytes internalize neutrophils in the bone marrow. That would stimulate megakaryocytes to increase platelet production [30,31], explaining why the platelet count increased on day 2 through day 7.

Due to operation and management aspects for triage during day 1 and day 2, collection of blood samples and CBC data from victims/patients will not be available in reality. Nevertheless, these samples and their data should be present during day 3, and changes in RBCs could be used to distinguish the victims/patients having been exposed to nothing, Hemo, RI, or CI. The question of whether the severity of CBC depletion caused by RI or CI can be prorated to estimate the radiation dose requires further exploration. In the case of radiation combined with skin wound trauma, the impact of CI on severity of mortality was proportional to the size of the skin wound [32]. However, a longitudinal analysis of total WBC and differential leukocyte counts of nonhuman primates after total body irradiation [33] indicated that the slowly worsening effect of CI on leukocyte depletion probably presents an increased triage time window in certain scenarios, such as mass casualty events.

Members of the scientific community interested in responses to irradiation combined with hemorrhage should be aware that this report provides preliminary data obtained from a mouse model that may be insightful in an emergency to evaluate human victims suffering from radiological accidents or nuclear detonation. These data must next be validated in minipigs or nonhuman primates to investigate if a similar change can be found and confirmed. Nevertheless, it should be kept in mind that if only one human blood sample was available to be examined, and if the differences in WBCs or RBCs were caused by RI or CI, which were within a normal range of lower and upper limits, then these differences could be ignored/diluted and not helpful for triage. Therefore, other parameters, such as IL-1β, IL = 6, IL-17A, TNF-α [6], Fms-related tyrosine kinase 3 (Flt3) ligand [26], or miR-34a [9], should be concurrently measured as well for confirmation.

It is important to keep in consideration that sex-based disparity to radiosensitivity, age, and subsidiary diseases, such as hypertension, high cholesterol, diabetes, infection, etc., can potentially impact these outcomes [9]. Further investigations are required in this line.

The study findings provide potential implications for emergency response strategies and disaster management in scenarios involving CI. The care providers should aggressively treat the CI patients and observe them closely for any minute changes, because CI vastly enhances the RI-induced ARS. Thus, these patients must be managed more carefully. 

In terms of the feasibility of implementing such an approach in real-word emergency settings, logistical challenges and resource constraints should be kept in mind. Measuring CBCs from victims’ blood samples could be completed very rapidly using newly FDA-approved point of care devices (in as little as 5 min), although sequential CBCs would be needed. Measuring cytokines as mentioned above using ELISA kits could be challenging due to a lack of availability of a portable microplate reader, and each read would require at least 2 h of processing, although many samples could be read on one plate. Advanced development of assay technology in this regard is still a pressing need. 

## 4. Materials and Methods

### 4.1. Ethics Statement

A facility accredited by the Association for Assessment and Accreditation of Laboratory Animal Care—International (AAALACI) was used to perform the research project. Animals and their proposed procedures were under review and approval by the Institutional Animal Care and Use Committee (IACUC) of the Armed Forces Radiobiology Research Institute.

### 4.2. Animals and Experimental Design

Male CD2F1 mice (10 weeks old) were purchased from Harlan Labs (Indianapolis, IN, USA). They were acclimated to their surroundings for 14 days before beginning the study. They were randomly placed in cages with 8 mice per cage at a temperature of 68–75ºF in a light-controlled room with a 12 h light–dark cycle. They were randomly divided into four experimental groups with N = 6/group: sham (0 Gy), hemorrhage (Hemo, 20% total blood volume), Radiation Injury (RI), and RI + Hemo. After Hemo, RI, or RI + Hemo, mice were placed into clean cages with 2–4 mice per cage. Proper food (standard rodent chow, Harlan Teklad 8604, Inotiv, Gaithersburg, MD, USA) and acidified water were provided to mice *ad libitum*. The health status of each animal was monitored and recorded daily according to the approved IACUC protocol. 

### 4.3. Gamma Irradiation

Mice were placed in well-ventilated acrylic restrainers and exposed to single whole-body 8.75 Gy ^60^Co γ-photon radiation at a dose rate of approximately 0.6 Gy/min in a ^60^Co facility (Nordion Inc., Otawa, ON, Canada) at AFRRI. Dosimetry was performed using the alanine/electron paramagnetic resonance system. Calibration of the dose rate with alanine was traceable to the National Institute of Standards and Technology and the National Physics Laboratory of the United Kingdom. Sham-irradiated mice were placed in the same acrylic restrainers, taken to the radiation facility, and restrained for the time required for irradiation. We selected 8.75 Gy for the biomarker elucidation, because this dose has previously been studied with this mouse strain in the literature [6,34].

### 4.4. Hemorrhage (Hemo)

Within 2 h post-RI, mice were anesthetized under isoflurane (~3%) plus 97% oxygen and bled 0% (Sham, RI) or 20% (Hemo, CI) of total blood volume via the submandibular vein, as previously described [35]. Briefly, the jaw of the anesthetized mouse was cleaned with a 70% EtOH wipe, and glycerol was applied to the surface of the jaw to allow for ease of collection and measurement of blood loss. A 5 mm Goldenrod animal lancet (MEDIpoint, Inc.; Mineola, NY, USA) for facial vein blood samples was used to puncture the submandibular vein of the mouse and heparinized hematocrit collection tubes (75 mm; Drummond Scientific Co.; Broomall, PA, USA) were marked and used to collect the appropriate amount of blood to ensure 20% of total blood volume was extracted during the hemorrhage process. The volume of blood collected was based upon each individual mouse’s body mass [36]. Euthanasia was conducted according to the recommendations and guidelines of the American Veterinary Medical Association at the end of each specific time point.

### 4.5. Assessment of Blood Cell Profile in Peripheral Blood

For CBC studies, mice at specific endpoints were placed under anesthesia through isoflurane inhalation for the entire period of blood collection. After blood collection, animals were immediately euthanized through a confirmatory cervical dislocation.

Blood samples were collected in EDTA tubes 30 d after RI or CI and assessed with the ADVIA 2120 Hematology System (Siemens, Deerfield, IL, USA). Differential analysis was conducted using the peroxidase method and the light scattering techniques recommended by the manufacturer.

### 4.6. Statistical Analysis

Data are expressed as means ± SEM (N = 6 per group per time point). ANOVA, Bonferroni’s inequality test, and Student’s *t-* test were used for comparison of groups. For all data, statistical significance was accepted at *p* < 0.05.

## 5. Conclusions

In a mass casualty event, Sham, Hemo alone, RI alone, or CI with both RI + Hemo can happen simultaneously or sequentially. Dynamic changes in differential CBC can be a potential tool to distinguish these single injuries from CI using WBCs for the early time points and RBCs for the later time points post-RI and post-CI. For hemorrhage alone, neutrophil counts at 4–5 h and platelets for day 2 through day 7 can be used as tools or for confirmation. 

## Figures and Tables

**Figure 1 ijms-25-02988-f001:**
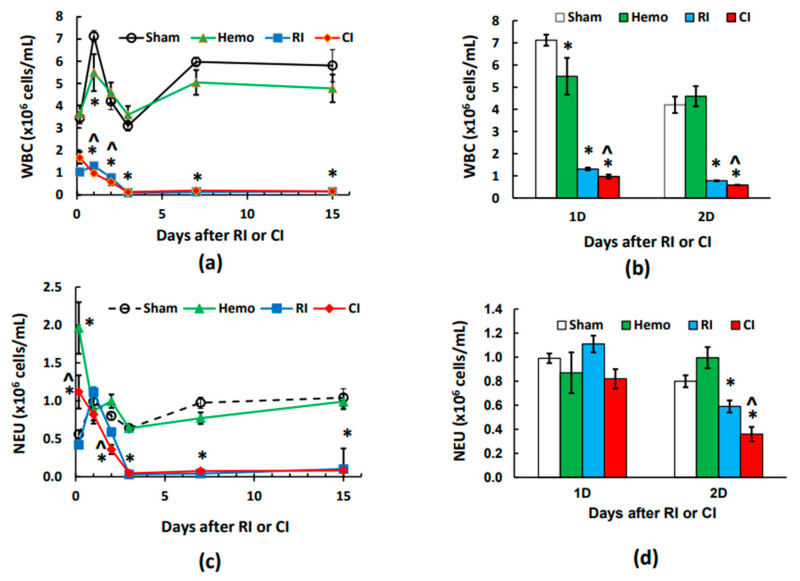
Effects of radiation alone or combined radiation injury with hemorrhage on white blood cells and neutrophils. Animals were exposed to 8.75 Gy alone or followed by 20% hemorrhage. Complete blood cells were counted in blood collected in sham mice, Hemo alone mice, radiation alone mice, or radiation + Hemo mice at 4–5 h and on days 1, 2, 3, 7, and 15. Data are mean ± sem with N = 6 per group per time point. (**a**) WBC throughout 15 days; (**b**) WBCs on day 1 and day 2; (**c**) neutrophils throughout 15 days; (**d**) neutrophils on day 1 and day 2. * *p* < 0.05 vs. sham group; ^ *p* < 0.05 vs. sham, Hemo, and RI at the specific time point. Hemo: 20% hemorrhage; RI: 8.75 Gy; CI: 8.75 Gy + 20% hemorrhage; WBC: white blood cells; NEU: neutrophils; D: day.

**Figure 2 ijms-25-02988-f002:**
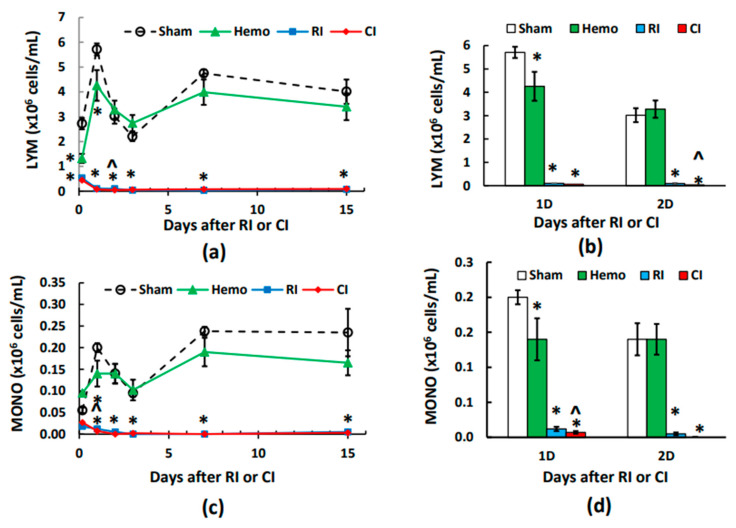
Effects of radiation alone or combined radiation injury with hemorrhage on lymphocytes and monocytes. Animals were exposed to 8.75 Gy alone or followed by 20% hemorrhage. Complete blood cells were counted in blood collected in sham mice, Hemo alone mice, radiation alone mice, or radiation + Hemo mice at 4–5 h and on days 1, 2, 3, 7, and 15. Data are mean ± sem with N = 6 per group per time point. (**a**) Lymphocytes throughout 15 days; (**b**) lymphocytes on day 1 and day 2; (**c**) monocytes throughout 15 days; (**d**) monocytes on day 1 and day 2. * *p* < 0.05 vs. sham group; ^ *p* < 0.05 vs. sham, Hemo, and RI at the specific time point. Hemo: 20% hemorrhage; RI: 8.75 Gy; CI: 8.75 Gy + 20% hemorrhage; LYM: lymphocytes; MONO: monocytes; D: day.

**Figure 3 ijms-25-02988-f003:**
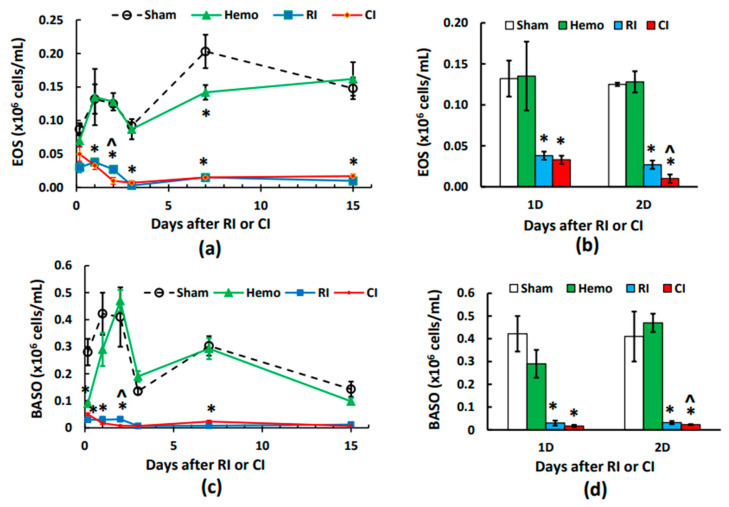
Effects of radiation alone or combined radiation injury with hemorrhage on eosinophils and basophils. Animals were exposed to 8.75 Gy alone or followed by 20% hemorrhage. Complete blood cells were counted in blood collected in sham mice, Hemo alone mice, radiation alone mice, or radiation + Hemo mice at 4–5 h and on days 1, 2, 3, 7, and 15. Data are mean ± sem with N = 6 per group per time point. (**a**) Eosinophils throughout 15 days; (**b**) eosinophils on day 1 and day 2; (**c**) basophils throughout 15 days; (**d**) basophils on day 1 and day 2. * *p* < 0.05 vs. sham group; ^ *p* < 0.05 vs. sham, Hemo, and RI at the specific time point. Hemo: 20% hemorrhage; RI: 8.75 Gy; CI: 8.75 Gy + 20% hemorrhage; EOS: eosinophils; BASO: basophils; D: day.

**Figure 4 ijms-25-02988-f004:**
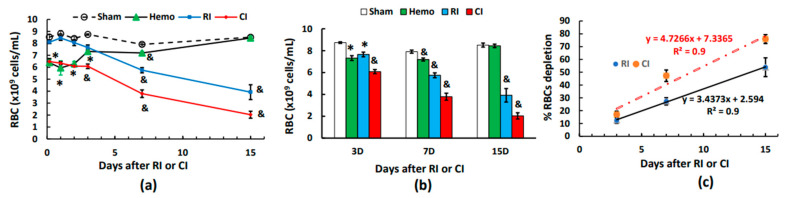
Effects of radiation alone or combined radiation injury with hemorrhage on red blood cells. Animals were exposed to 8.75 Gy alone or followed by 20% hemorrhage. Complete blood cells were counted in blood collected in sham mice, Hemo alone mice, radiation alone mice, or radiation + Hemo mice at 4–5 h and on days 1, 2, 3, 7, and 15. Data are mean ± sem with N = 6 per group per time point. (**a**) Red blood cells throughout 15 days; (**b**) red blood cells on days 1, 2, and 3; (**c**) red blood cells on days 3, 7, and 15 with slopes for RI and CI, respectively. * *p* < 0.05 vs. sham group; & *p* < 0.05 vs. the rest of the 3 groups at the specific time point. Hemo: 20% hemorrhage; RI: 8.75 Gy; CI: 8.75 Gy + 20% hemorrhage; RBC: red blood cells; D: day; R^2^: correlation coefficient.

**Figure 5 ijms-25-02988-f005:**
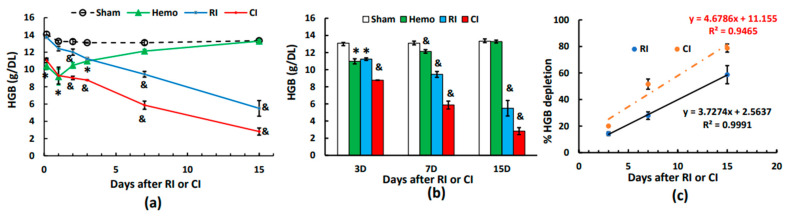
Effects of radiation alone or combined radiation injury with hemorrhage on hemoglobin levels. Animals were exposed to 8.75 Gy alone or followed by 20% hemorrhage. Complete blood cells were counted in blood collected in sham mice, Hemo alone mice, radiation alone mice, or radiation + Hemo mice at 4–5 h and on days 1, 2, 3, 7, and 15. Data are mean ± sem with N = 6 per group per time point. (**a**) Hemoglobin levels throughout 15 days; (**b**) hemoglobin levels on days 1, 2, and 3; (**c**) hemoglobin levels on days 3, 7, and 15 with slopes for RI and CI, respectively. * *p* < 0.05 vs. sham group; & *p* < 0.05 vs. the rest of the 3 groups at the specific time point. Hemo: 20% hemorrhage; RI: 8.75 Gy; CI: 8.75 Gy + 20% hemorrhage; HGB: hemoglobin; D: day; R^2^: correlation coefficient.

**Figure 6 ijms-25-02988-f006:**
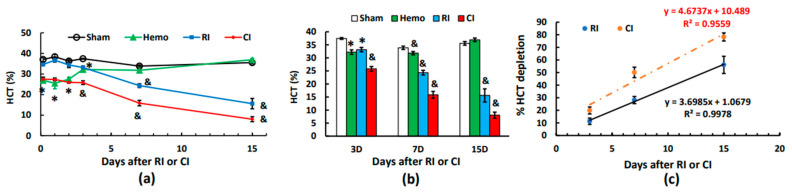
Effects of radiation alone or combined radiation injury with hemorrhage on hematocrit readings. Animals were exposed to 8.75 Gy alone or followed by 20% hemorrhage. Complete blood cells were counted in blood collected in sham mice, Hemo alone mice, radiation alone mice, or radiation + Hemo mice at 4–5 h and on days 1, 2, 3, 7, and 15. Data are mean ± sem with N = 6 per group per time point. (**a**) Hematocrit readings throughout 15 days; (**b**) hematocrit readings on days 1, 2, and 3; (**c**) hematocrit readings on days 3, 7, and 15 with slops for RI and CI, respectively. * *p* < 0.05 vs. sham group; & *p* < 0.05 vs. the rest of the 3 groups at the specific time point. Hemo: 20% hemorrhage; RI: 8.75 Gy; CI: 8.75 Gy + 20%; HCT: hematocrits; D: day; R^2^: correlation coefficient.

**Figure 7 ijms-25-02988-f007:**
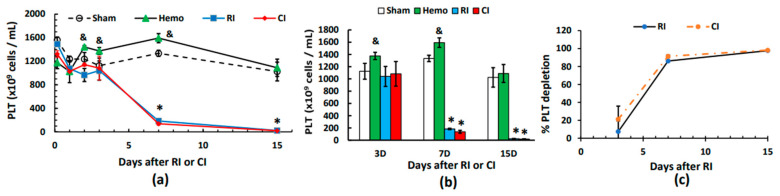
Effects of radiation alone or combined radiation injury with hemorrhage on platelets. Animals were exposed to 8.75 Gy alone or followed by 20% hemorrhage. Complete blood cells were counted in blood collected in sham mice, Hemo alone mice, radiation alone mice, or radiation + Hemo mice at 4–5 h and on days 1, 2, 3, 7, and 15. Data are mean ± sem with N = 6 per group per time point. (**a**) Platelets throughout 15 days; (**b**) platelets on days 1, 2, and 3; (**c**) platelets on days 3, 7, and 15 with a non-linear relationship for RI and CI, respectively. * *p* < 0.05 vs. sham group; & *p* < 0.05 vs. the rest of the 3 groups at the specific time point. Hemo: 20% hemorrhage; RI: 8.75 Gy; CI: 8.75 Gy + 20% hemorrhage; PLT: platelets; D: day.

**Table 1 ijms-25-02988-t001:** Relative complete blood count (CBC) changes to respective sham groups at different time points.

Cell Type	Condition	4–5 h	1D	2D	3D	7D	15D
WBCs	Hemo	−	↓	−	−	−	−
	RI	↓	↓	↓	↓	↓	↓
	CI	↓	↓↓	↓↓	↓	↓	↓
NEU	Hemo	↑	−	−	−	−	−
	RI	↓	−	↓	↓	↓	↓
	CI	↑	−	↓↓	↓	↓	↓
LYM	Hemo	↓	↓	−	−	−	−
	RI	↓	↓	↓	↓	↓	↓
	CI	↓	↓	↓↓	↓	↓	↓
MONO	Hemo	↑	↓	−	−	−	−
	RI	↓	↓	↓	↓	↓	↓
	CI	↓	↓↓	↓	↓	↓	↓
EOS	Hemo	−	−	−	−	↓	−
	RI	↓	↓	↓	↓	↓	↓
	CI	↓	↓	↓↓	↓	↓	↓
BASO	Hemo	↓	−	−	−	−	−
	RI	↓	↓	↓	↓	↓	↓
	CI	↓	↓	↓↓	↓	↓	↓
RBCs	Hemo	↓	↓	↓	−	−	−
	RI	−	−	−	↓	↓	↓
	CI	↓	↓	↓	↓↓	↓↓	↓↓
HGB	Hemo	↓	↓	↓	↓	−	−
	RI	−	−	↓	↓	↓	↓
	CI	↓	↓	↓↓	↓↓	↓↓	↓↓
HCT	Hemo	↓	↓	↓	↓	−	−
	RI	−	−	−	↓	↓	↓
	CI	↓	↓	↓	↓↓	↓↓	↓↓
PLT	Hemo	↓	−	↑	↑	↑	−
	RI	−	−	↓	−	↓	↓
	CI	↓	−	−	−	↓	↓

Hemo: 20% hemorrhage; RI: radiation at 8.75 Gy; CI: 8.75 Gy + 20% hemorrhage; h: hour; WBCs: white blood cells; NEU: neutrophils; LYM: lymphocytes; MONO: monocytes; EOS: eosinophils; BASO: basophils; RBCs: red blood cells; HGB: hemoglobin; HCT: hematocrit; PLT: platelets; D: day; −: no change; ↑: increase; ↓: decrease; ↓↓: further decrease.

## Data Availability

The data presented in this study are available from the authors upon reasonable request.

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
