# Peer review of "Effects of Hemorrhage on Hematopoietic Cell Depletion after a Combined Injury with Radiation: Role of White Blood Cells and Red Blood Cells as Biomarkers"

_ijms, 2024, doi:10.3390/ijms25052988_

Round 1

Reviewer 1 Report

Comments and Suggestions for Authors

The paper is concise, interesting and important. From the language and technical point of view everything is Ok, except minor issues. But on the basis of the presented data I do not agree with the conclusions. According to me you can clearly distinguish between Sham/Hemo and RI / CI, but there is no big difference between RI and CI, opposite to what is stated in the paper. The difference in WBC for early time point and RBS for later time point is visible (hardly, but I believe it really is present) when you compare groups of mice. But if You have only 1 blood sample to examine, the difference will be diluted in the inter individual differences of WBC or RBS and therefore not useful for the triage. I would draw such conclusion from the data presented in the paper. There is no reference to possible  inter individual differences of blood counts and its influence for the minute difference found between RI and CI. Should be discussed.

Line 48

iNOS – explanation?

Line 64

CBC – explanation – it is explained afterwards as counts of blood cells, should be explain here.

Line 65

If you skip the middle of the sentence than You get “Mice … are relevant to human biology” – I would rephrase this sentence somehow?

Line 253

will be Could?

Line 258

Maybe: total and differential leukocyte counts of non…..?

Line 257

The sentence is not understandable for me: what effect?

Line 272

Space between day 7.

Line 318

To big font of half of the line?

Author Response

Responses to reviewer 1’s suggestions and comments

Thank you for your constructive comments that made this manuscript at a better level. We had incorporated your suggestions/comments into the revised version. Your comments are addressed one by one. The changed areas are highlighted in yellow for your efficient review.

  1. The paper is concise, interesting, and important. From the language and technical point of view everything is Ok, except minor issues. But on the basis of the presented data I do not agree with the conclusions. According to me you can clearly distinguish between Sham/Hemo and RI / CI, but there is no big difference between RI and CI, opposite to what is stated in the paper. The difference in WBC for early time point and RBS for later time point is visible (hardly, but I believe it really is present) when you compare groups of mice. But if You have only 1 blood sample to examine, the difference will be diluted in the inter individual differences of WBC or RBS and therefore not useful for the triage. I would draw such conclusion from the data presented in the paper. There is no reference to possible inter individual differences of blood counts and its influence for the minute difference found between RI and CI. Should be discussed.

Answers: Thank you for recognizing the paper being concise, interesting, and important, and the language and technical point of view are OK. The difference in WBC for early time points and RBS for later time points is indeed visible and statistically significant without any doubts in a mouse model. We agree with you that under a mass casualty, when human blood samples were examined, the small but statistically significantly elevated value can be lost due to comparing to the normal range of human data defined with lower limit and upper limit. Therefore, we propose to combine other parameters along for verification. Please see lines 284-288.

  1. Line 48: iNOS – explanation?

Answers: inducible nitric oxide synthase is included. Please see line 50.

  1. Line 64: CBC – explanation – it is explained afterwards as counts of blood cells, should be explain here.

Answers: It has been corrected. Please see lines 65-66.

  1. Line 65: If you skip the middle of the sentence than You get “Mice … are relevant to human biology” – I would rephrase this sentence somehow?

Answers: It has been corrected. Please see lines 66-68.

  1. Line 253: will be Could?

Answers: It has been corrected. Please see line 270.

  1. Line 258: Maybe: total and differential leukocyte counts of non…..?

Answers: It has been corrected. Please see line 275.

  1. Line 257: The sentence is not understandable for me: what effect?

Answers: Sorry for the confusion. It has been rewritten for its clarification. Please see lines 276-278.

  1. Line 272: Space between day 7.

Answers: It has been corrected. Please see line 310.

  1. Line 318: To big font of half of the line?

Answers: It has been deleted. Please see lines 355-356.

Reviewer 2 Report

Comments and Suggestions for Authors

International Journal of Molecular Sciences (Manuscript ID: ijms-2889140), Comments to the Authors:

Title: Effects of hemorrhage on hematopoietic cell depletion after a combined injury with radiation: Role of white blood cells and red blood cells as biomarkers

Comments

The manuscript provides a thorough investigation into the effects of the of RI or CI on blood cell depletion as a biomarker to differentiate the two. Male CD2F1 mice were exposed to 8.75 Gy γ- radiation (60Co). Within 2 h of RI, animals were bled under anesthesia 0% (RI) or 20% (CI) of total blood volume. Blood samples were collected at 4-5 h and days 1, 2, 3, 7, and 15 after RI. CI decreased WBC at 4-5 h and continued to decrease them until day 3; counts then stayed at the nadir up to day 15. CI decreased neutrophils, lymphocytes, monocytes, eosinophils and basophils more than RI on day 1 or day 2. CI decreased RBCs, hemoglobin and hematocrit on days 7 and 15 more than RI, whereas hemorrhage alone returned to the baseline on days 7 and 15. RBCs depleted after CI faster than post RI. Hemorrhage alone increased platelet counts on days 2, 3, and 7, which returned to the baseline on day 15. The data suggested that WBC depletion may be a potential biomarker within 2 days post-RI and -CI and RBC depletion after 3 days post-RI and -CI.

I think the submitted manuscript can be accepted for publication after the authors respond to the following comments:

1.     The study could benefit from a more in-depth discussion on the limitations and potential confounding factors that may have influenced the outcomes.

2.     The implications of the study findings for clinical practice could be more explicitly articulated, providing clearer guidance on the practical applications of the research.

3.     The study could explore additional mechanistic insights into the molecular pathways involved in hematopoietic cell depletion to enhance the understanding of the underlying biological processes.

4.     The study's focus on blood cell depletion as the primary outcome measure may overlook other relevant endpoints that could provide a more comprehensive assessment of the combined injury effects.

5.     The discussion section could further elaborate on the potential implications of the study findings for emergency response strategies and disaster management in scenarios involving combined injuries.

6.     While the temporal dynamics of differential blood counts are extensively discussed, the mechanisms underlying these changes are not sufficiently explored. Providing insights into the physiological processes driving the observed alterations in WBCs, RBCs, and platelets would enhance the scientific rigor of the discussion.

7.     The proposed utility of differential CBC for triage in mass casualty incidents lacks practical considerations. Discussing the feasibility of implementing such an approach in real-world emergency settings, including logistical challenges and resource constraints, would improve the applicability of the findings to clinical practice.

Author Response

Responses to Reviewer 2’s comments and suggestions

Thank you for your constructive comments that made this manuscript at a better level. We had incorporated your suggestions/comments into the revised version. Your comments are addressed one by one. The changed areas are highlighted in yellow for your efficient review.

  1. The study could benefit from a more in-depth discussion on the limitations and potential confounding factors that may have influenced the outcomes.

Answers: Yes, we sincerely appreciated your suggestions. Please see lines 252-266 and 283-304. For confounding factors, please see lines 289-291.

  1. The implications of the study findings for clinical practice could be more explicitly articulated, providing clearer guidance on the practical applications of the research.

Answers: We added them accordingly. Please see lines 292-296.

  1. The study could explore additional mechanistic insights into the molecular pathways involved in hematopoietic cell depletion to enhance the understanding of the underlying biological processes.

Answers: It has been inserted accordingly. Please see lines 252-266.

  1. The study's focus on blood cell depletion as the primary outcome measure may overlook other relevant endpoints that could provide a more comprehensive assessment of the combined injury effects.

Answers: It has been inserted accordingly. Please see lines 257-259.

  1. The discussion section could further elaborate on the potential implications of the study findings for emergency response strategies and disaster management in scenarios involving combined injuries.

Answers: It has been inserted accordingly. Please see lines 292-296.

  1. While the temporal dynamics of differential blood counts are extensively discussed, the mechanisms underlying these changes are not sufficiently explored. Providing insights into the physiological processes driving the observed alterations in WBCs, RBCs, and platelets would enhance the scientific rigor of the discussion.

Answers: It has been added accordingly. Please see lines 252-266.

  1. The proposed utility of differential CBC for triage in mass casualty incidents lacks practical considerations. Discussing the feasibility of implementing such an approach in real-world emergency settings, including logistical challenges and resource constraints, would improve the applicability of the findings to clinical practice.

Answers: Thank you for the suggestion. It has been discussed accordingly. Please see lines 297-304.
